# EAT–*Lancet* Recommendations and Their Viability in Chile (2014–2023): A Decade-Long Cost Comparison Between a Healthy and Sustainable Basket and the Basic Food Basket

**DOI:** 10.3390/nu17121953

**Published:** 2025-06-08

**Authors:** Daniel Egaña Rojas, Patricia Gálvez Espinoza, Lorena Rodríguez-Osiac, Francisco Cerecera Cabalín

**Affiliations:** 1Departamento de Atención Primaria y Salud Familiar, Facultad de Medicina, Universidad de Chile, Av. Independencia 1027, Santiago 8380453, Chile; degana@uchile.cl; 2Departamento de Nutrición, Facultad de Medicina, Universidad de Chile, Av. Independencia 1027, Santiago 8380453, Chile; pagalvez@uchile.cl; 3Escuela de Salud Pública, Facultad de Medicina, Universidad de Chile, Av. Independencia 1027, Santiago 8380453, Chile; lorenarodriguez@uchile.cl; 4Facultad de Agronomía y Sistemas Naturales, Pontificia Universidad Católica de Chile, Avenida Vicuña Mackenna 4860, Santiago 6904411, Chile

**Keywords:** affordability, diet, health, nutrition, poverty, public policy, sustainability

## Abstract

Background/Objectives: Addressing the global syndemic of obesity, undernutrition, and climate change requires a shift toward healthy and sustainable diets. This study examines the feasibility and cost implications of implementing a Healthy and Sustainable Basic Food Basket in Chile that aligns with the EAT–*Lancet* diet recommendations, through its comparison with the current Basic Food Basket used for the poverty line definition. Methods: The Healthy and Sustainable Basic Food Basket was constructed based on the EAT–*Lancet* dietary model and was uniquely adapted to reflect the observed consumption patterns of Chile’s lowest income quintile, allowing for a more realistic affordability assessment for vulnerable populations. Food prices from the National Institute of Statistics were analyzed over a 10-year period (2014–2023). Results: This study found that the Healthy and Sustainable Basic Food Basket provides 2001 kcal per day with a balanced macronutrient distribution. However, its average cost was 13.9% higher than the Basic Food Basket, posing a significant economic barrier for low-income populations. The cost gap varied seasonally, peaking in October (21.1% higher) and narrowing in December (4.6% higher). Long-term trends showed increasing costs for both baskets, with the Healthy and Sustainable Basic Food Basket reaching its highest price in 2023, further limiting affordability. Conclusions: These findings highlight the limitations of current poverty measurement frameworks in Chile, which prioritize caloric sufficiency over nutritional quality and sustainability. This suggests a need for policy revisions to incorporate the cost of healthy and sustainable diets into poverty assessments and social protection programs. Key policy recommendations include promoting healthier diets and improved food nutrition, supporting low-carbon foods, regulating local food production and supply systems, and encouraging seasonal, local consumption. This study underscores the need for structural interventions to ensure equitable access to sustainable diets, addressing both public health and environmental concerns.

## 1. Introduction

The global syndemic of obesity, undernutrition, and climate change—as highlighted by *The Lancet* in 2019 [1]—underscores the urgent need for sustainable and healthy dietary patterns. The report highlighted that although the initial focus was on obesity, the three interconnected issues share common drivers and collectively threaten human health and survival. Therefore, effective actions to combat any of these three pandemics would simultaneously impact the others [1].

Globally, an estimated 42% of the population is affected by overnutrition, a figure projected to rise to 51% by 2035 [2]. Additionally, 390 million adults and approximately 200 million children experience some degree of undernutrition [3]. In terms of climate change, data from 2019 indicate that CO_2_, CH_4_, and N_2_O concentrations have been at their highest levels in thousands of years, with visible impacts worldwide [4] and particularly in Latin America and the Caribbean [5]. These three global pandemics, overnutrition, undernutrition, and climate change, are fundamentally linked to food systems both as contributing factors and as potential solutions.

The high prevalence of overnutrition in Chile, affecting over 70% of adults and 50% of school-aged children [6,7], highlights the inadequacy of current dietary patterns and the potential role of food policy in promoting healthier options. In contrast, low weight among adults and undernutrition among children remain below 3% of the population [6,7]. Moreover, overnutrition has been strongly associated with poor dietary habits. According to the latest National Food Consumption Survey, 95% of the population requires dietary modifications [8], with 28.6% of total caloric intake derived from ultra-processed foods [9].

This inadequate diet not only impacts human health but also threatens planetary sustainability. The significant environmental impact of the global food system, responsible for 34% of total greenhouse gas emissions [10], is mirrored in Chile, where the average diet results in a daily per capita carbon footprint of 4.67 kg of CO_2_ and a water footprint of 4177 L [11]. These substantial environmental burdens underscore the crucial need to incorporate sustainability considerations into the national food policy. In this context, the sustainability of diets has become a pressing global health necessity [12].

A critical food security concern is that processed foods of low nutritional quality and high caloric density often have a lower cost [1]. This makes it crucial to determine the cost of a healthy and sustainable diet as a viable alternative, particularly as Chile’s poorest households are the most affected by rising living costs and food prices.

The Basic Food Basket (BFB) has been developed as an attempt to establish an objective criterion for measuring poverty. Since the work of economist Mollie Orshansky in the 1960s, efforts have been made to create a framework linking a diet with minimally sufficient nutritional standards to the economic measurement of poverty [13,14]. Following this approach, in the 1970s and within the framework of a project by the Economic Commission for Latin America and the Caribbean (ECLAC), Oscar Altimir proposed using a BFB that would cover minimum nutritional requirements as a comparative measure of the poverty line [15]. Based on this methodology, in the following decade, ECLAC developed a comparative estimation across different Latin American countries [16], leading to the first poverty measurements in Chile [17,18].

The construction of the first BFB in Chile was based on data from the Fourth Household Budget Survey, conducted in 1987 [19]. This survey included household food expenditures, allowing for an estimate of apparent food consumption. The selected reference group was the third income quintile since the apparent consumption of the first two quintiles was below the minimum caloric intake threshold. The most representative foods in this quintile’s total food expenditure were identified [17]. Next, the quantity of each food item was estimated using consumer prices reported by the National Institute of Statistics (INE), and finally, the caloric contribution of the BFB was determined using the Chilean Food Composition Table. This version of the BFB provided an average of 2176 daily calories [17]. As a result, an individual was classified as living in extreme poverty if their monthly income fell below the cost of the BFB. Likewise, they were considered below the poverty line (“poor”) if their per capita income was lower than twice the monthly cost of the BFB [17].

In 2014, a new methodology for measuring poverty was introduced, adjusting previous parameters. The BFB was reformulated using the expenditure of the lowest income quintile according to the Seventh Household Budget Survey since this segment met the minimum nutritional requirements [20]. This revised BFB comprised 76 food items, providing an average of 2054 daily calories [21]. The updated version excluded alcoholic beverages (which had been included in the previous version) and reduced the availability of products high in saturated fats, added sugars, and energy density [21]. To determine the poverty line, an Orshansky coefficient of 2.68 was applied, with extreme poverty set at two-thirds of the poverty threshold [20]. According to the latest National Socioeconomic Characterization Survey [22], 6.5% of Chile’s population lives in poverty or extreme poverty based on income levels, meaning they lack the financial capacity to purchase sufficient basic food supplies.

While Orshansky initially proposed poverty thresholds based on nutritious diets, in Latin America and Chile, the BFB implementation has primarily focused on caloric sufficiency, with only minor adjustments to improve diet quality. In 2015, the Chilean Ministry of Health (MINSAL), the Pan American Health Organization (PAHO), and ECLAC commissioned the development of indicators to monitor the socioeconomic impact of non-communicable diseases, including the construction of a Quality Food Basket (QFB) [23]. The QFB was designed as a modification of the BFB to align with the Chilean Dietary Guidelines. However, the cost of the QFB was 38.7% higher than the BFB [23], and it did not account for the environmental sustainability of food consumption.

In 2019, *The Lancet* supported the creation of the EAT Commission, bringing together 37 scientists from 16 countries to “develop global scientific targets for healthy diets and sustainable food production” [24]. The commission focuses on two primary factors: healthy diets and sustainable food production. EAT–*Lancet* defines a healthy and nutritionally sufficient diet based on natural food groups, adapted to local food production and consumption patterns [24]. Additionally, this diet aligns with a planetary health perspective on food systems, designed to meet the objectives of the Paris Agreement by reducing greenhouse gas emissions [24].

Overall, the EAT–*Lancet* diet recommendations are largely consistent with Chile’s national dietary guidelines (Guías Alimentarias para Chile, [25]), with one notable exception: dairy consumption. While the EAT–*Lancet* diet recommends a sustainable daily intake of 250–500 mL of dairy products, Chilean guidelines advise at least three servings per day (600 mL) [25]. While historical factors may partially explain this difference [26,27], some experts, such as Marion Nestle, have suggested that industry influence may also play a role [28].

However, a significant research gap in the Chilean context is the absence of a comprehensive assessment of a healthy and sustainable dietary model, such as that proposed by the EAT–*Lancet* Commission, that is adapted to the actual food consumption patterns of the vulnerable population segment. Previous national studies on healthier food baskets did not systematically combine sustainability criteria with this adaptation to the dietary habits of lower-income households, nor did they typically conduct a longitudinal cost comparison of such an adapted basket against the official Basic Food Basket.

The overall goal of this study is to examine the feasibility and cost implications of implementing a Healthy and Sustainable Basic Food Basket (HSBFB) in Chile that aligns with the EAT–*Lancet* diet recommendations through its comparison with the current Basic Food Basket (BFB) used by Chile’s Ministry of Social Development for the poverty line definition. To achieve this goal, the specific objectives are to (1) construct such an HSBFB, aligned with EAT–*Lancet* recommendations; (2) price this HSBFB; and (3) conduct a longitudinal comparison of this HSBFB against the BFB.

## 2. Materials and Methods

### 2.1. Construction of the Healthy and Sustainable Basic Food Basket

To construct an HSBFB, the dietary model proposed by the EAT–*Lancet* Commission [24] was used as a reference. This model establishes recommended intake ranges per food group, expressed in grams per day. The EAT–*Lancet* diet proposes eight food groups. These food groups are whole grains (such as rice, wheat, maize, and others); tubers (such as potatoes and cassava); vegetables; fruits; dairy products (whole milk or equivalents); protein sources (including beef, lamb, and pork; poultry; eggs; fish; legumes; and nuts and seeds); added fats (such as unsaturated oils, saturated fats); and added sugars (Table 1).

To determine the specific food items within each group, a methodology similar to the one used for constructing the BFB was applied:The apparent consumption (purchases) of the lowest income quintile was identified using data from the Ninth Household Budget Survey [29].The most representative food items within each group were selected, based on their share of total consumption in this income segment.The recommended gram intake per food group (as per EAT–*Lancet*) was then distributed proportionally according to the observed consumption pattern.

For example, in the case of vegetables, a total of 25 different types of vegetables (out of 28) were recorded. The proportion of each vegetable in the total quantity purchased by the lowest income quintile was determined. Following this method, the most consumed vegetable was tomato (22.57%), followed by onion (13.25%), lemons (11.07%), and so on, down to asparagus (0.13%). This percentage distribution was then applied to the total amount recommended by the EAT–*Lancet* diet for vegetables (300 g per day). Thus, tomatoes, representing 22.57% of total vegetable intake, translated to 67.72 g per day.

After constructing the EAT–*Lancet*-based diet, which mirrors the structure of healthy food consumption within the lowest income quintile, the nutritional values of each food were calculated—including total calories, protein, carbohydrates, and fats. These calculations were based on data from the USDA FoodData Central database [30].

Additionally, a Food Transformation Index (FTI) was applied to each food item. This index accounts for modifications (loss or expansion) that a food undergoes between its natural, unprocessed state and its prepared state, excluding non-edible parts due to cultural or biological factors [31]. Two reference sources were used for this adjustment: a national source as the primary reference [32] and an international reference [33]. The application of the FTI to foods modifies the calculation of the amount of food an individual needs to purchase to meet recommended quantities. When a food item has an FTI below 1, it signifies a loss during preparation (e.g., peeling and discarding the skin of a vegetable). Therefore, a larger quantity of that food must be purchased to satisfy the recommended intake. Conversely, an FTI greater than 1 indicates a weight gain during preparation (e.g., rice, dry pasta, and dried beans, which absorb water during cooking). Consequently, to achieve a specific gram intake, a smaller quantity of such food is required. As a result, for example, the 300 g of vegetables recommended by the EAT–*Lancet* diet transformed into 369.84 g purchased in the market after applying the FTI.

### 2.2. Food Pricing and Data Collection

Each selected food item was matched to its corresponding entry in the dataset of the National Institute of Statistics (INE), which records monthly food prices for cost calculation purposes. The price series used was sourced from the Consumer Price Index (CPI) series [34], which updates its representative food tracking portfolio every five years. Since these are consumer prices, those measured in kilograms or the most common market size are considered to avoid distorting the retail market price.

Potential values outside the data’s admissible range are identified based on reference guides and databases. These cases can be observed because the original information series was constructed with the aim of monitoring the Consumer Price Index (CPI), where different prices could be weighted in various ways, beyond their current value. This methodology was applied to the CPI series base years 2009, 2013, and 2018, standardizing the reported food items across these periods and linking the data to construct a continuous price series spanning from 2010 to 2023, covering 229 food price points.

Before determining the average market price and minimum observed price for each month, extreme values were removed from the dataset. The correction of outliers was performed by cross-referencing the data with other reliable national sources, such as those provided by the Office of Agricultural Studies and Policies (ODEPA) of the Ministry of Agriculture [35]. This adjustment ensures that prices remain within the typical range for local urban markets, reducing dispersion in the data matrix. The average prices were obtained by collapsing the general average of standardized prices by month, year, and product. For minimum prices, each food item was identified per month, and the lowest observed price was selected.

### 2.3. Cost Calculation and Comparative Analysis Framework

The 2010–2023 price series covers the majority of the 168-month period, although some months lacked recorded data. In these cases, missing prices were estimated using the nearest-neighbor method within the same period, following the established literature on hierarchical clustering techniques [36].

From the 2010–2023 price series, only the prices from April 2014 (when the current BFB measurement began in Chile) to December 2023 were used, covering a total of 128 months.

For the purposes of this study, the minimum price was used to calculate the cost of the HSBFB, as this is the price basis used to construct the BFB [20]. After constructing the price series and determining the monthly cost of the HSBFB, these values were compared with the official monthly BFB price reported by the Ministry of Social Development [37].

Finally, the cost of both baskets was converted to U.S. dollars to facilitate international comparisons and contextualize the economic implications. To achieve this, the daily observed exchange rate for the entire period (April 2014 to December 2023) was obtained from the Central Bank of Chile (Banco Central de Chile, 2024 [38]) and averaged for each month.

## 3. Results

### 3.1. Composition and Nutritional Profile of the Healthy and Sustainable Basic Food Basket

The HSBFB consists of 66 commonly consumed food items within the lowest income quintile, as identified in the Household Budget Survey. It includes 25 types of vegetables, 23 types of fruits, and 10 protein sources, in addition to added sugars, fats, cereals, and starchy tubers (Table 2). The prominence of plant-based foods and the inclusion of diverse protein sources within these items are key features aligning this basket with the EAT–*Lancet* recommendations for health and sustainability.

The HSBFB provides an energy intake of 2001 kilocalories per day, with 14.71% derived from protein, 32.36% from lipids, and the remainder from carbohydrates (see Table 3).

### 3.2. Historical Cost Comparison in Local Currency

In April 2014, the HSBFB cost CLP 39,193 per month (USD 70.70 at the time), an 11.9% increase over the BFB, which was CLP 35,019 (USD 63.10) that same month. This cost difference remained relatively stable over the years. By April 2023, the HSBFB had reached CLP 70,731 (USD 88) per month, 10.5% higher than the BFB, which stood at CLP 64,741 (USD 79.70) for the same period (see Figure 1).

Despite the long-term stability in price differences, seasonal variations in cost disparities between the two baskets were observed throughout the year. In certain months, the HSBFB cost approaches the BFB, while in others, the gap widens. Analyzing data from April 2014 to December 2023, the largest cost differences occurred primarily between August and October, reaching a peak in October 2017, when the HSBFB was 21.1% more expensive than the BFB. Conversely, the smallest price differences tend to occur between December and February, reaching the lowest point in December 2018, when the HSBFB was only 4.6% more expensive than the BFB. Over 128 months, the HSBFB remained 13.9% more expensive than the BFB, underscoring a structural cost barrier that limits access to healthier diets for lower-income populations.

### 3.3. Historical Cost Comparison in U.S. Dollars

In terms of cost expressed in daily USD, both baskets remained relatively stable until October 2022. In April 2014, the BFB cost USD 2.10 per day, rising slightly to USD 2.14 per day in October 2022, with an average cost of USD 2.07 during that period. The HSBFB followed a similar trend, costing USD 2.36 per day in April 2014 and reaching USD 2.52 per day in October 2022, with an average cost of USD 2.36 throughout the period (see Figure 2).

However, from November 2022 to December 2023, both the BFB and HSBFB experienced a significant cost increase compared to the previous eight years. During this period, the BFB reached its highest price in February 2023, peaking at USD 2.68 per day, with an average cost of USD 2.55 per day. Similarly, the HSBFB reached its highest cost in March 2023, peaking at USD 3.11 per day, with an average cost of USD 2.86 per day for this period.

Finally, the overall average cost from April 2014 to December 2023 was USD 2.13 per day for the BFB and USD 2.43 per day for the HSBFB. These long-term price trends reveal a persistent affordability gap between the BFB and HSBFB. The next section examines these disparities in the context of global food policy and dietary affordability research.

## 4. Discussion

### 4.1. Interpretation of Findings and Alignment with Dietary Standards

This study presents an estimation of the cost of a healthy and sustainable diet in Chile, analyzed longitudinally over nearly 10 years on a monthly basis. The diet constructed in this analysis aligns with the HSBFB model proposed by the EAT–*Lancet* Commission.

The diet aligns with energy sufficiency standards in Latin America, where basic food baskets range between 1987 and 2141 kilocalories per day [39]. Similarly, the number of food items included (66) is consistent with the quantity identified in other basic food baskets in the region [39].

Unlike previous efforts to construct a healthy food basket for Chile [23], this proposal incorporates sustainability considerations, reducing the consumption of animal-based protein while increasing the intake of legumes, in line with EAT Commission recommendations and previous analyses of Chilean dietary patterns [11].

### 4.2. Comparison with International Dietary Cost Studies

Studies using 2011 data show that a nutritionally adequate diet costs, on average, 2.66 times more than one that meets only caloric sufficiency [40], increasing from USD 0.57 to USD 1.35 per day. However, these assessments do not consider dietary diversity or sustainability.

When compared with similar dietary cost studies, the price of the EAT–*Lancet* diet in Chile is consistent with findings in the literature. Hirvonen et al. conducted a global affordability analysis of the EAT–*Lancet* diet across 159 countries, and they determined that, on average, this diet cost USD 2.84 per day (IQR 2.41–3.16) in 2011 [41]. While our data do not extend to 2011, the average cost for the entire study period (April 2014 to December 2023) is USD 2.43 per day. The same study also indicated that the average cost of a diet focusing solely on nutrient adequacy (without considering planetary sustainability) was USD 1.60 per day (IQR 1.41–1.78), which is notably lower than the USD 2.13 per day observed for the BFB in Chile during the study period.

Additionally, a recent study by Herforth et al. used 2017 data to estimate the cost of a Healthy Diet Basket (HDB) based on Food-Based Dietary Guidelines (FBDGs). Their analysis, covering over 159 countries, estimated an average daily cost of USD 3.32, which is 36% higher than the USD 2.43 average cost of the HSBFB in Chile for 2017 [42], highlighting cross-country variability in dietary cost estimates and food prices.

### 4.3. Understanding Variability in Diet Cost Comparisons

Country-specific studies from New Zealand, Brazil, Argentina, Mexico, and Australia have compared healthy diets to standard national diets [43,44,45,46,47,48,49], yielding mixed results. Studies in New Zealand, Argentina, and Denmark concluded that healthy diets are between 3.9% and 31.7% more expensive [46,48,49]. In contrast, research from Australia, Mexico, and Brazil suggests that a healthy diet is actually cheaper than the typical diet in those countries [43,44,45,47]. These discrepancies highlight the importance of considering local food systems and dietary patterns when formulating policies aimed at promoting healthy and sustainable diets.

The discrepancy in these findings can be attributed to multiple factors. Jensen et al. [48] attributed the higher cost of a healthy diet in Denmark to the inclusion of organic and Nordic-sourced products, which tend to be more expensive than those in the standard diet. For New Zealand and Argentina, the higher cost was linked to the higher price per calorie of fruits and vegetables compared to ultra-processed, energy-dense foods [46,49,50]. On the other hand, in Australia, Brazil, and Mexico, the lower cost of healthy diets was attributed to reduced consumption of red meat and ultra-processed foods, as well as the inclusion of affordable fruits and vegetables [43,44,45,47]. In Australia, the exemption of fresh produce from taxation has also contributed to lower costs [49], reinforcing healthy public policies.

Two key methodological factors explain the differences between our findings and previous studies. Most of the reviewed studies [43,44,45,47,49] construct healthy diets using the DIETCOST tool developed by INFORMAS [51] and follow either EAT–*Lancet* recommendations or Food-Based Dietary Guidelines (FBDGs) from each country. In contrast, our study adheres to EAT–*Lancet* guidelines but retains the observed consumption structure of the lowest income quintile from the Ninth Household Budget Survey [29], reflecting real-world consumption patterns in Chile.

A second methodological difference concerns the reference diet. The reviewed studies compare healthy diets to the “average current diet” [44,46,48,49] or the “typical diet” of each country [45,47]. These comparisons incorporate total population consumption, including higher-income groups. In contrast, our study defines the reference diet as the BFB, which was constructed based on the spending patterns of the lowest income quintile, meaning it reflects the calorically sufficient diet of Chile’s poorest population [20].

### 4.4. Policy Implications for Poverty Measurement and Social Equity

This study has several policy implications. The finding that a healthy and sustainable diet is more expensive than a diet meeting only caloric sufficiency, as defined by the BFB, calls into question the adequacy of the current poverty line. Under existing conditions, maintaining social reproduction above the poverty line does not guarantee a healthy life for a significant portion of the population, effectively perpetuating forms of malnutrition. Moreover, by failing to meet sustainability criteria, lower-income populations lack the financial resources to mitigate the environmental impact of their food choices—a consequence of an inequitable food system [52].

Given that the HSBFB is 13.9% more expensive than the current BFB, policymakers should consider revising poverty thresholds to better reflect the cost of a healthy and sustainable diet. Durán and Kremerman [53] simulated the impact of the Quality Food Basket [23] on the poverty line, indicating that a 36.1% increase in food costs would raise income poverty levels from 8.6% to 19.8% nationwide. Updating this analysis based on the HSBFB is one of the key research challenges arising from this study.

Furthermore, the findings regarding the HSBFB align strongly with international policy recommendations aimed at tackling the global syndemic structurally. Burgaz et al. emphasize the critical need for integrated public policies that simultaneously address multiple dimensions of food systems, including nutrition, environmental sustainability, and social equity [54,55,56]. Among the most urgent policy actions identified as relevant to this study are measures to improve the affordability of healthier and more sustainable diets, provide subsidies for healthier and more sustainable foods, and offer incentives or subsidies for sustainable production methods (crops, livestock, and fish) [54,56]. According to Burgaz et al. [56], implementing such interconnected policies is essential for addressing the underlying drivers of malnutrition, obesity, and climate change from a structural, population-based perspective. Applied in the Chilean context, these types of interventions could directly address the affordability barrier identified for the HSBFB, helping to lower its cost relative to the BFB, improve physical access and availability for vulnerable populations, and ultimately mitigate dietary and social inequalities.

Another policy implication of this study concerns the variability of dietary costs. As demonstrated by the findings, the overall cost variation of the BFB is similar to that of the HSBFB (see Figure 1 and Figure 2). This suggests that the price tendency of a fresh food-based healthy diet is similar to a basket that includes processed and ultra-processed foods, which supports the state promotion of an HSBFB. Naturally, HSBFB prices exhibit greater seasonality, with lower costs observed in summer months (lowest annual cycle cost), but they remain stable over the long term. This seasonal fluctuation could be mitigated by promoting the consumption of seasonal and locally produced foods.

### 4.5. Limitations and Strengths of This Study

One limitation of this study concerns the comparison diet used against the EAT–*Lancet* diet. Unlike most reviewed studies, which compare against the average national diet, this study relies on the Household Budget Survey and the BFB, which is constructed based on the minimum consumption necessary to meet 2000 kilocalories [20]. Although this methodological choice complicates direct comparisons with other studies, it could also be seen as a more rigorous and replicable approach.

On the other hand, this study’s main strength lies in the methodology used to construct the HSBFB. Unlike other studies that propose healthy and sustainable diets through computational simulations, this research utilizes real-world food consumption patterns observed in Chilean society, specifically within the lowest income quintile. This ensures that the foods included in the proposed basket are culturally acceptable within Chilean society.

## 5. Conclusions

This study confirms the feasibility of constructing a Healthy and Sustainable Basic Food Basket (HSBFB) for Chile, aligned with EAT–*Lancet* recommendations and local consumption patterns. However, the 13.9% higher cost of the HSBFB compared to the current Basic Food Basket (BFB) presents a significant affordability barrier for low-income populations, showing the need to re-evaluate the current poverty measurement frameworks and targeted policy interventions. Our findings highlight the urgent need to incorporate nutritional quality and environmental sustainability into poverty measurement frameworks, ensuring policies that support equitable access to sustainable diets. To ensure broader accessibility and overall viability of sustainable diets, policymakers should implement policies that actively promote nutritious, low-carbon foods, and foster more equitable and transparent food system dynamics to reduce disparities. To reduce the cost gap between the BFB and HSBFB, targeted policies—such as incentives for sustainable food production and seasonal and local food consumption—must be prioritized. These policies could include facilitating access to healthy diets and improved food nutrition, as well as strategies that promote local production, encourage seasonal food consumption, and mitigate price seasonality. Such an approach would be crucial in advancing toward more equitable and sustainable food systems, contributing to both public health and the achievement of sustainable development goals. Future research should focus on evaluating the real-world effectiveness and scalability of the proposed policy interventions, such as those aimed at fostering healthy food environments and strengthening local production, in improving the affordability and uptake of the HSBFB across diverse low-income communities in Chile.

## Figures and Tables

**Figure 1 nutrients-17-01953-f001:**
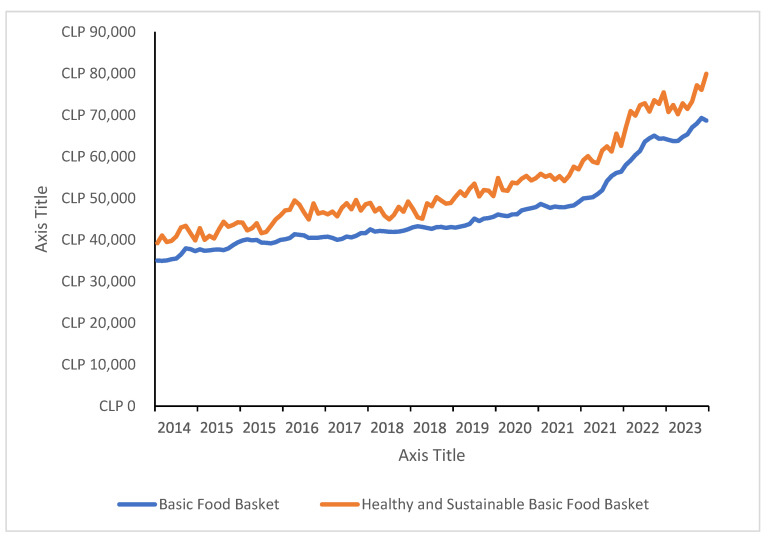
Historical comparison between the BFB and HSBFB in Chilean pesos (monthly).

**Figure 2 nutrients-17-01953-f002:**
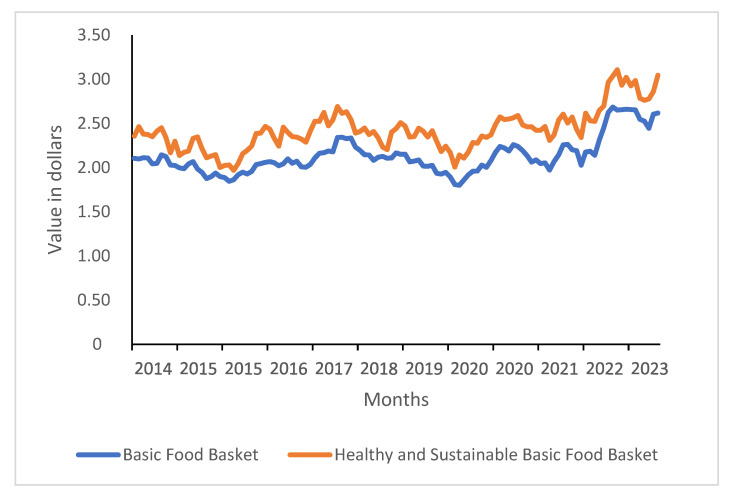
Historical comparison between the BFB and HSBFB in U.S. dollars (daily).

**Table 1 nutrients-17-01953-t001:** Comparison of foods in the Basic Food Basket and the Healthy and Sustainable Basic Food Basket by EAT–*Lancet* diet food groups and subgroups.

Food Group (EAT–*Lancet*)	Subgroup	Food Items in the Healthy and Sustainable Basic Food Basket	Food Items in the Basic Food Basket ^1^
Whole Grains	Rice, Wheat, Corn, and Others	Rice; Bread; Spaghetti	Rice; Bread; Spaghetti; Oats
Tubers or Starchy Vegetables	Potatoes and Cassava	Potato	Potato
Vegetables	All Types of Vegetables	Tomatoes; Onion; Lemon; Carrot; Avocado; Corn; Lettuce; Sweet Pumpkin; Bell Pepper; Chard; Zucchini; Cucumber; Green Beans; Garlic; Beetroot; Spinach; Cauliflower; Artichoke; Mushrooms; Asparagus	Lettuce; Zucchini; Squash; Lemon; Avocado; Tomato; Carrot; Onion; Corn
Fruits	All Types of Fruits	Banana; Apple; Orange; Peach and Nectarines; Grapes; Mandarins; Strawberry; Melons; Pears; Kiwi; Watermelon; Cherry; Plum; Prickly Pear; Cucumber; Mango; Apricot; Grapefruit; Cherimoya	Banana; Orange; Grape; Plum; Peach; Apple
Dairy Products	Whole Milk or Equivalents	Whole Milk	Whole Milk; Whole Milk Powder; Yogurt; Gouda Cheese; Fresh Cheese; Cream Cheese
Protein Sources	Beef, Lamb, and Pork	Ground Beef; Pork Chop; Black Beef; Pork Pulp; Pork Ribs	Rump Cap; Eye of Round; Ground beef; Pork Chop; Pork Ribs; Boneless Pork; Boneless Lamb; Frankfurter; Sausage; Pork Ham
	Chicken and Other Poultry	Chicken Thigh; Chicken Breast; Whole Chicken; Turkey Thigh; Turkey Breast; Ground Turkey	Ground Turkey; Chicken Breast; Whole Chicken; Chicken Leg; Poultry Sausage
	Eggs	Egg	Egg
	Fish	Hake; Pomfret; Salmon	Hake; Mussels; Jack Mackerel
	Legumes	Dried Beans; Lentils	Dried Beans; Lentils
	Nuts and Seeds		Salted Peanuts
Added Fats	Unsaturated Oils	Vegetable Oil; Olive Oil	Vegetable Oil
	Saturated Oils	Butter	Butter; Margarine
Added Sugars	All Types of Sugars	Sugar	Sugar
Outside the EAT–*Lancet* Food Groups			Sweet Cookie; Savory Cracker; Cake; Pre-Made Pizza Base; Wheat Flour; Chocolate; Candy; Ice Cream; Tomato Sauce; Coffee; Tea; Mineral Water; Soft Drink; Energy Drink; Isotonic Drink; Fruit Juice; Fruit Nectar; Powdered Soft Drink; Hot Dog; French Fries; Sweet Pastries; Soup; Fast Food Combo; Avocado Toast; Ham and Cheese Sandwich; Roasted Chicken; Savory Filled Pastries.

^1^ Foods contained in the current Basic Food Basket used by Chile’s Ministry of Social Development for poverty line measurement.

**Table 2 nutrients-17-01953-t002:** Grams of consumption and grams after food transformation, by groups and subgroups in the Healthy and Sustainable Basket.

Food Group	Subgroup	Food	Grams to Consume (g)	Adjustment Factor	Grams with FTI (g) ^1^
Whole Grains	Rice, Wheat, Corn, and Others	Rice	28.14	2.5	11.26
		Bread	204.33	1	204.33
		Spaghetti	17.52	2.36	7.43
	Total Subgroup		250.00		223.02
Tubers or Starchy Vegetables	Potatoes and Cassava	Potato	100.00	1.04	96.15
	Total Subgroup		100.00		96.15
Vegetables	All Types of Vegetables	Tomatoes	67.72	0.83	81.59
		Onion	39.77	0.78	50.99
		Lemon	33.22	0.58	57.27
		Carrot	28.17	0.94	29.97
		Avocado	26.86	0.73	36.80
		Corn	26.68	1.04	25.65
		Lettuce	25.77	1	25.77
		Sweet Pumpkin	17.19	0.7622	22.55
		Bell Pepper	5.84	0.86	6.79
		Chard	5.28	1.0192	5.18
		Zucchini	4.87	0.931	5.23
		Cucumber	4.82	0.76	6.34
		Green Beans	2.79	0.98	2.85
		Garlic	2.69	0.87	3.09
		Beetroot	2.19	0.7812	2.81
		Spinach	1.88	0.97	1.94
		Cauliflower	1.80	0.93	1.94
		Artichoke	1.12	1.17	0.96
		Mushrooms	0.94	0.62	1.52
		Asparagus	0.40	0.6405	0.62
	Total Subgroup		300.00		369.85
Fruits	All Types of Fruits	Banana	86.41	0.66	130.92
		Apple	54.84	0.83	66.07
		Orange	37.73	0.73	51.69
		Peach and Nectarines	22.59	0.75	30.12
		Grapes	18.34	0.92	19.93
		Mandarins	16.89	0.66	25.59
		Strawberry	14.06	0.92	15.28
		Melons	13.95	0.51	26.83
		Pears	11.71	0.85	13.78
		Kiwi	6.50	0.82	7.92
		Watermelon	5.14	0.52	9.88
		Cherry	3.58	0.84	4.27
		Plum	2.33	0.95	2.45
		Prickly Pear	1.67	0.95	1.76
		Cucumber	1.64	0.85	1.93
		Mango	1.44	0.71	2.03
		Apricot	0.53	0.95	0.56
		Grapefruit	0.40	0.68	0.59
		Cherimoya	0.25	1	0.25
	Total Subgroup		300.00		411.85
Dairy Products	Whole Milk or Equivalents	Whole Milk	400.00	1	400.00
	Total Subgroup		400.00		400.00
Protein Sources	Beef, Lamb, and Pork	Ground Beef	4.99	0.6	8.32
		Pork Chop	3.37	0.5609	6.00
		Black Beef	2.70	0.64	4.21
		Pork Pulp	1.95	0.73	2.67
		Pork Ribs	1.00	0.286	3.49
	Total Subgroup		14.00		24.69
	Chicken and Other Poultry	Chicken Thigh	17.81	0.5544	32.12
		Chicken Breast	8.33	0.5928	14.04
		Whole Chicken	2.55	0.507	5.02
		Turkey Thigh	0.17	0.672	0.25
		Turkey Breast	0.12	0.79	0.15
		Ground Turkey	0.03	0.79	0.04
	Total Subgroup		29.00		51.63
	Eggs	Egg	15.00	0.9944	15.08
	Total Subgroup		15.00		15.08
	Fish	Hake	14.98	0.3871	38.69
		Pomfret	11.18	0.4543	24.61
		Salmon	1.84	0.89	2.07
	Total Subgroup		28.00		65.37
	Legumes	Dried Beans	67.60	1.4	48.28
		Lentils	32.40	1.6	20.25
	Total Subgroup		100.00		68.54
	Nuts and Seeds ^2^	Nuts and Seeds	0.00	0	0.00
	Total Subgroup		0.00		0.00
Added Fats	Unsaturated Oils	Vegetable Oil	19.55	1	19.55
		Olive Oil	0.45	1	0.45
	Total Subgroup		20.00		20.00
	Saturated Oils	Butter	11.80	1	11.80
	Total Subgroup		11.80		11.80
Added Sugars	All Types of Sugars	Sugar	31.00	1	31.00
	Total Subgroup		31.00		31.00

^1^ “Grams with FTI” (purchased) calculated as “Grams to Consume”/Adjustment Factor. The Adjustment Factor quantifies weight changes during food preparation (see Section 2 for details); factors derived from Cáceres and Lataste (2021) [32] and Pecot and Watt (1975) [33]. ^2^ Set to zero as the consumption reported for nuts and seeds by the lowest income quintile in the Household Budget Survey (INE, 2024a [29]) was marginal, and thus not included in the allocation for this adapted basket model.

**Table 3 nutrients-17-01953-t003:** Energy contribution by the food subgroups of the Healthy and Sustainable Basket.

Food Group	Subgroup	Energy (kcal)	Protein (g)	Carbohydrates (g)	Lipids (g)
Whole Grains	Rice, Wheat, Corn, and Others	711.46	22.14	117.95	15.61
Tubers or Starchy Vegetables	Potatoes and Cassava	76.00	2.06	16.30	0.25
Vegetables	All Types of Vegetables	136.55	4.44	18.77	4.87
Fruits	All Types of Fruits	178.18	2.41	40.65	2.04
Dairy Products	Whole Milk or Equivalents	256.00	13.20	18.80	14.80
Protein Sources	Beef, Lamb, and Pork	29.21	3.58	0.00	1.59
	Chicken and Other Poultry	49.14	7.77	0.02	1.85
	Eggs	22.35	1.86	0.05	1.64
	Fish	34.14	6.55	0.00	0.67
	Legumes	131.55	9.47	16.86	0.40
	Nuts and Seeds	0.00	0.00	0.00	0.00
Added Fats	Unsaturated Oils	165.85	0.00	0.00	18.44
	Saturated Oils	88.97	0.07	0.08	9.82
Added Sugars	All Types of Sugars	121.83	0.03	30.75	0.00
	TOTAL	2001.22	73.58	260.24	71.97
Percentage of Calories per Macronutrient			14.71%	52.02%	32.36%

## Data Availability

Data supporting the findings of this study, specifically concerning the methodology and results related to price construction, are available from https://doi.org/10.34691/UCHILE/WUUIDL.

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
