# Peer review of "EAT–Lancet Recommendations and Their Viability in Chile (2014–2023): A Decade-Long Cost Comparison Between a Healthy and Sustainable Basket and the Basic Food Basket"

_nutrients, 2025, doi:10.3390/nu17121953_

Round 1

Reviewer 1 Report

Comments and Suggestions for Authors

Nutrients

Manuscript Draft

Manuscript Number: 3635836

Title: EAT-Lancet Recommendations and Their Viability in Chile (2014–2023): A Decade-Long Cost Comparison Between a Healthy and Sustainable Basket and the Basic Food Basket

Article Type: Research article

General Comments on MDPI Questions that Reviewers must answer:

  • Is the manuscript clear, relevant, and well presented? 

The manuscript requires improvements in concept clarification as well as results presentation. The analysis conducts a recent ten-year historical time series of two different consumer market baskets in Chile and assesses the viability of both baskets in terms of affordability. However, in order for this manuscript to be published in MDPI Nutrients, there are the following FIVE major substantive edits/clarifications that need to be made:

1) It is extremely important when you are presenting time series data to show both real and nominal prices (this pertains to both graphs in Figure 1 and Figure 2). Since the current time series are in inflation-adjusted real values, please also present these graphs in nominal terms either on the same graph OR in separate graphs as Figure A1 and Figure A2 in Appendix A inserted between the Conclusions/Back Matter sections and the References.

2) Please add a new Table 1 in the Methods section where you contrast both types of food baskets along critical attributes that distinguish them. This will add contextual depth for the reader to enhance their understanding.

3) The Food Transformation Index (FTI) used in the research methodology requires more in-depth explanation since the logic of the adjustment is NOT intuitive. For example for rice with a FTI = 2.5, how can the final version weigh 11.26 grams after boiling where it absorbs water that is added to the initial 28.14 grams of dry grain? For an avocado initially weighing 26.86 grams with an FTI = 0.73, is the final consumed product of 38.6 grams due to other ingredients being added to make value-added guacamole? There needs to be an in-depth paragraph added in the Methods section where such examples are clearly justified and explained to the reader. It would be beneficial if these explanations are added as a last column to Table 1.

4) Please add how the nuance to own-price elasticity of demand for different food types (e.g., necessary staples versus luxuries) to the Discussion section citing literature on this. For example, if the more nutritious Healthy and Sustainable Basket (HSB) foods have relatively more elastic own-price demand elasticity, lower income consumers would be more price responsive (e.g., decrease demand more drastically with unit increases in price) for these luxury foods. This is driven by the a) income effect and the b) substitution effect. Higher prices of the HSB versus the Basic Food Basket (BFB) means consumers have less disposable income to spend on other things if they choose the BFB over the HSB. It is reasonable to assume that lower income consumers are “substituting” BFB for HSB due to the higher price of this more sustainable food basket and from what it sounds like, they are in essence “priced out” of consuming HSB which is more sustainable than the BFB. This presents a classical conundrum which needs to be fleshed out in the Discussion section as its own sub-section.

5) Please apply and evaluate economic recommendations for dealing with 4) above in the Discussion section. For example, would the Chilean government or local, national, or international NGO’s be in a position to subsidize low-income consumers so that they can better afford the HSB?

Please also make the following FIFTEEN minor edits:

1) Delete the words Background/Objectives: and Methods: and Results: from the Abstract as MDPI does not follow this format.

2) Do NOT use abbreviations in the Abstract.

3) Change L39 to:

    Keywords: affordability; diet; health; nutrition; poverty; public policy; sustainability

4) Paragraphs by definition are 3 sentences (1 topic sentence followed a minimum of 2 supporting sentences). Please correct this in the manuscript such as on L91-94, etc. Either add more sentences or merge with another paragraph if the topic is the same and can be consolidated.

5) Expand the last paragraph by clearly stating the goal(s) and objective(s) of the research. Typically the objectives are numbered.

6) Write L138-146 in paragraph format and not in bullet point format.

7) Please use sub-headers where all major words are capitalized for the Methods, Results, and Discussion. For example for the Methods section: 1. Index Calculations; 3.2. Inflation and Price Adjustments

8) For Table 1 and Table 2, please a) delete all “g” and put “(g)” in the header so for example Grams to Consume (g) so the data is easier to understand, b) only the first header row needs to be in bold, c) widen the first 3 columns so there are no rows that take up two row spaces particularly the 3rd Food column.

9) Table and figure headers should be in line with the table (e.g., do not indent L216-217).

10) Table headers such as on L216-217 should only capitalize words that are typically capitalized (please edit this for all table and figure headers):

Table 1. Grams of consumption and grams after food transformation by groups and subgroups in the Healthy and Sustainable Basket.

11) For Figure 1 and Figure 2, please a) remove the grey box around the graph, b) add black x-axis and y-axis lines, c) change the $ symbol in Figure 1 for the y-axis labels to the symbol for Chilean pesos, d) remove the grey horizontal lines, e) change the $- at the bottom of the y-axis to a numerical value, f) change the y-axis labels to the year format (e.g., 2014, 2015, etc.) in a horizontal orientation and not an upright vertical orientation that is very difficult to read and understand, and g) write out BFB and HSBFB in the legend.

12) The Conclusion section should be merged into one paragraph. Add to the end of this merged paragraph a couple of sentences on what future research can do to improve upon the present work.

13) Author Contributions on L449-451 need to follow the exact format and categories of the Word template on the MDPI Nutrients website under Instructions for Authors.

14) Delete (APC) on L452 as this is not used at all in what follows. Thus the abbreviation is redundant.

15) The formatting of the References need to follow the exact format provided in the MDPI Nutrients Word template. For example, the ISO4 abbreviations for the journal on L492 this is Lancet Planet. Health which needs to be in italics and please also note that for journal articles, year needs to be in bold with the volume(issue) in italics (e.g., Lancet Planet. Health 2018, 2, e451–e461.) and note that there is a period after the page range and not a comma. The non-journal article citations need to follow the exact detail and formatting of the examples given in the MDPI Nutrients Word template.

  • Are the cited references mostly recent publications within the last 5 years? Are citations relevant? Are there an excessive number of self-citations?

There are 32 of 56 citations have been published recently since 2019. References are relevant and there are no excessive self-citations.

  • Is the manuscript scientifically sound? Does the experimental design appropriately test the hypothesis?

The manuscript needs to be clarified based on my FIVE major prior edits/clarifications. The goal and objectives of the research need to be listed in the last paragraph of the Introduction section (this is the norm across most academic journals).

  • Are the results reproducible based on the methods section?

The manuscript’s results are mostly reproducible based on what is described in 2. Methods. However, further written clarification to the writing on the Food Transformation Index is critically needed as already mentioned.

  • Are the figures and tables OK? Is the data analysis and interpretation valid?

Both of these questions were already addressed in my previous comments and edits.

  • Are conclusions consistent with the results presented?

The conclusions are consistent with the results with only minor edits such as adding two sentences to the end of the Conclusion section on future improvements.

  • Please evaluate the data availability and conflict of interest statements for adequacy.

Both of these Back Matter sections are fine.

Author Response

Dear Reviewer,

We sincerely thank you for your thoughtful and detailed feedback. Your comments have been instrumental in improving the clarity, depth, and rigor of our manuscript.

We have carefully considered each of your suggestions and have worked diligently to address your concerns. Your feedback has been invaluable in improving the quality of our manuscript, and we have made the necessary revisions to ensure that your points have been thoroughly incorporated.

We hope that the changes we have implemented meet your expectations and adequately respond to the issues you raised.

Kind regards,

Reviewer 2 Report

Comments and Suggestions for Authors

The thesis has a clear direction, but it also has some weaknesses. First, the introduction sufficiently explains the need for this study, but how does this study differ from previous research? What does it offer the reader, both academically and practically?

The authors state that the diet consists of 8 food groups. Please provide a reference or reason to support this.

The results are presented in a series of tables, but none of them are particularly statistically significant. They can be viewed as a simple report.

In the Discussion section, recent literature should be presented to justify the findings of this study, and similarities and differences based on previous research should be presented.

Author Response

(The authors gave the same response as above.)

Reviewer 3 Report

Comments and Suggestions for Authors

The paper requires deep review before accepting for publication.

  1. The aim presented in introduction and in abstract is different. It should be the same.
  2. The research gap is nit clear.
  3. The results should be better described.
  4. The paper should contain policy implications.
  5. Cost comparison between a healthy and sustainable basket is interesting. But the Authors should explain what is Sustainability issue.   Mainly this term concerns sustainable development of regionu, sektorem, farms and enterprises. The term sustainable basket is not well recognized in the literature.
Comments on the Quality of English Language

Can be a little improved.

Author Response

(The authors gave the same response as above.)

Round 2

Reviewer 1 Report

Comments and Suggestions for Authors

Nutrients

Manuscript Draft

Manuscript Number: 3635836

Title: EAT-Lancet Recommendations and Their Viability in Chile (2014–2023): A Decade-Long Cost Comparison Between a Healthy and Sustainable Basket and the Basic Food Basket

Article Type: Research article

Thank you for making requested edits and clarifications. Please also make the following SEVEN minor edits:

1) Sub-headers need to have a blank row above and NOT be indented as well as all sub-headers need to be in italics and have the second number followed by a period. The correct format for this was followed on L444-445 for the blank row but not the writing which should be: 5. Limitations and Strengths of the Study

2) Regarding 1) above, please do NOT indent and please put into italics on L152 (also delete the (HSBFB) since you are not supposed to use abbreviations in sub-headers. etc., L222 (add blank row above), L240, L261 (delete (CLP) since abbreviation not needed), L301 (add blank row above and delete (USD) since redundant), L342, L355 (add blank row above), L374 (add blank row above), L407 (add blank row above),

3) L213-221 needs to be one paragraph.

4) Add another supporting sentence to two-sentence paragraph on L241-244.

5) Delete the word see on L244

6) For Figure 1 and Figure 2, change the font color to black from grey.

7) The formatting of the References still need to follow the exact format provided in the MDPI Nutrients Word template for certain citations. For example, references on L494-495, L516-525, L531, L536-538, and L544-545 do not have enough information provided for the page proof editor to easily find let alone anyone reading the manuscript. Please use the following formats for cited works other than journal articles:

  1. Author 1, A.; Author 2, B. Title of the chapter. In Book Title, 2nd ed.; Editor 1, A., Editor 2, B., Eds.; Publisher: Publisher Location, Country, 2007; Volume 3, pp. 154–196.
  2. Author 1, A.; Author 2, B. Book Title, 3rd ed.; Publisher: Publisher Location, Country, 2008; pp. 154–196.
  3. Author 1, A.B.; Author 2, C. Title of Unpublished Work. Abbreviated Journal Name year, phrase indicating stage of publication (submitted; accepted; in press).
  4. Author 1, A.B. (University, City, State, Country); Author 2, C. (Institute, City, State, Country). Personal communication, 2012.
  5. Author 1, A.B.; Author 2, C.D.; Author 3, E.F. Title of Presentation. In Proceedings of the Name of the Conference, Location of Conference, Country, Date of Conference (Day Month Year).
  6. Author 1, A.B. Title of Thesis. Level of Thesis, Degree-Granting University, Location of University, Date of Completion.
  7. Title of Site. Available online: URL (accessed on Day Month Year).

Author Response

Dear Reviewer,

I would like to sincerely thank you for your thoughtful and constructive comments. Your suggestions have been extremely helpful in improving the quality and clarity of our manuscript. We truly appreciate the time and effort you dedicated to reviewing our work.

We have carefully addressed your comments from the second round of review, and we hope that the revisions represent a satisfactory advancement of the manuscript. We are also grateful for your consideration, which encourages us to believe that the manuscript is on a promising path toward publication.

Kind regards,

Reviewer 2 Report

Comments and Suggestions for Authors

The revision is now acceptable.

Well done~~

Author Response

Dear Reviewer,

I would like to sincerely thank you for your thoughtful and constructive comments. Your suggestions have been extremely helpful in improving the quality and clarity of our manuscript. We truly appreciate the time and effort you dedicated to reviewing our work.

We are also grateful for your consideration, which encourages us to believe that the manuscript is on a promising path toward publication.

Kind regards,

Reviewer 3 Report

Comments and Suggestions for Authors

The authors responded to my comments and mostly improved the paer. It can be published.

Author Response

(The authors gave the same response as above.)
